# Human Height Estimation by Color Deep Learning and Depth 3D Conversion

**Dong-seok Lee [1], Jong-soo Kim [2], Seok Chan Jeong [3]**  **and Soon-kak Kwon [1],\***

[1] Department of Computer Software Engineering, Dong-eui University, Busan 47340, Korea; ulsan333@gmail.com

[2] Software Convergence Center, Dong-eui University, Busan 47340, Korea; avantas@naver.com

[3] Department of e-Business, Convergence of IT Devices Institute, AI Grand ICT Research Center, Dong-eui University, Busan 47340, Korea; scjeong@deu.ac.kr

\* Correspondence: skkwon@deu.ac.kr; Tel.: +82-51-897-1727

**Abstract:** In this study, an estimation method for human height is proposed using color and depth information. Color images are used for deep learning by mask R-CNN to detect a human body and a human head separately. If color images are not available for extracting the human body region due to low light environment, then the human body region is extracted by comparing between current frame in depth video and a pre-stored background depth image. The topmost point of the human head region is extracted as the top of the head and the bottommost point of the human body region as the bottom of the foot. The depth value of the head top-point is corrected to a pixel value that has high similarity to a neighboring pixel. The position of the body bottom-point is corrected by calculating a depth gradient between vertically adjacent pixels. Two head-top and foot-bottom points are converted into 3D real-world coordinates using depth information. Two real-world coordinates estimate human height by measuring a Euclidean distance. Estimation errors for human height are corrected as the average of accumulated heights. In experiment results, we achieve that the estimated errors of human height with a standing state are 0.7% and 2.2% when the human body region is extracted by mask R-CNN and the background depth image, respectively.

**Keywords:** human-height estimation; depth video; depth 3D conversion; artificial intelligence; convolutional neural networks

## 1. Introduction

The physical measurements of a person such as human height, body width and stride length are important bases for identifying a person from video. For example, the height of the person captured by a surveillance video is important evidence for identifying a suspect. Physical quantities are also used as important information for continuously tracking a specific person in video surveillance system consisting of multiple cameras [1]. A specific behavior such as falling down can be recognized by detecting changes in human height. Various studies have been conducted to estimate human height from color video. Human height is estimated by obtaining 3D information of the human body from color video [2–6]. Both the position and the pose of the camera are required in order to obtain 3D information of human body. Human height can also be estimated by calculating the ratio of the length between human body and a reference object whose length is already known [7–12]. The estimation methods of human height based on color video have a disadvantage in that the camera parameters or information about a reference object are required.

Depth video stores depth values, meaning the distances between subjects and the camera. The pixels of depth video are converted to 3D coordinates by the depth values. Object detection [13–15]

and behavior recognition [16–18] by depth video are possible by extracting the 3D features of the objects. Recently, smartphones recognize a human face through an equipped TOF sensor for recognizing the identity of a person. Object lengths can be also measured from depth video without the additional information, so the problems of the human-height estimation based on color video can be solved by using depth video.

The field of artificial intelligence has made significant progress by researching neural network structures which consist of multilayers. In particular, convolutional neural network (CNN) [19] respectably improves object detection that categorizes the object and detects the boundary boxes and pixels of the objects [20–26].

In this study, a human-height estimation method is proposed using depth and color information. The human-height estimation is improved by extracting a human body and a human head from color information and by measuring human height from depth information. The human body and the human head of current frame in color video are extracted through mask R-CNN [26]. If color images are not available due to a low light environment, then the human body region is extracted by comparing between current frame in depth video and a pre-stored background depth image. The topmost point of the human head region is extracted as a head-top and bottommost point of the human body region as a foot-bottom. Two top head and foot-bottom points are converted to 3D real-world coordinates by these image coordinates and depth pixel values. Human height is estimated by calculating a Euclidean distance between two real-world coordinates.

The proposed method improves the human-height estimation by using both color and depth information and by applying mask R-CNN which is an art-of-state algorithm for object detection. In addition, the proposed method removes the need for the camera parameters or the length of other object in the human-height estimation using depth information.

This study is organized as follows: In Section 2, the related works for object detection by CNN and for the human-height estimation based on color or depth video are described. In Section 3, the human-height estimation by depth and color videos is proposed. The experimental results of the proposed method are presented in Section 4. Finally, a conclusion for this study is described in Section 5.

## 2. Related Works

### 2.1. Object Detection from Color Information by Convolutional Neural Network

Object detection problems in color image can generally be classified into four categories: classification, localization, detection and object segmentation. First, the classification determines an object category for single object in the image. Second, the localization finds the boundary box for single object in the image. Third, the detection finds the boundary boxes and determines object categories for multiple objects. Finally, the object segmentation finds pixels where each object is. CNN can solve whole categories of object detection problems. CNN replaces the weights of the neural networks with kernels which are rectangular filters. Generally, object detection through CNN are classified as 1-stage and 2-stage methods [27]. The 1-stage method performs both the location and the classification at once. The 2-stage method performs the classification after the location. The 1-stage method is faster than the 2-stage method but is less accurate. R-CNN [20] is a first proposed method for the detection through CNN. R-CNN applies a selective search algorithm to find the boundary box with a high probability where an object exists. The selective search algorithm is the method of constructing the boundary box by connecting adjacent pixels with similar texture, color and intensity. The object is classified through SVM (support vector machine). The feature map of the boundary boxes is extracted through AlexNet. R-CNN has disadvantages that the object detection is seriously slow and SVM should be trained separately from CNN. Fast R-CNN [21] has higher performance than R-CNN. Fast R-CNN applies a RoIPool algorithm and introduces a softmax classifier instead of SVM, so the feature map extraction and the classification are integrated into one neural network. faster R-CNN [22] replaces the

selective search algorithm into a region proposal network (RPN) so whole processes of object detection are performed in one CNN. YOLO [23,24] is the 1-stage method for object detection. YOLO defines the object detection problem as a regression problem. YOLO divides an input image into the grid cells of a certain size. The boundary boxes and the reliability of the object are predicted for each cell at same time. YOLO detects the objects more quickly than the 2-stage methods but is less accurate. SSD [25] allows the various sizes of the grid cells in order to increase the accuracy of object detection. mask R-CNN [26] is proposed for the object segmentation unlike other R-CNNs.

### 2.2. Object Length Measurement from Color or Depth Information

Length estimation methods based on color video are classified into length estimations by camera parameters [2–6], by vanishing points [7–12], by prior statistical knowledge [28,29], by gaits [30,31] and by neural networks [32,33]. The length estimation methods by the camera parameters generate an image projection model into an color image by the focal length, the height and the poses of a camera. The object length is estimated by converting the 2D coordinates in the pixels of the image into 3D coordinates through the projection model. The length estimation methods by the camera parameters have a disadvantage that the accurate camera parameters should be provided in advance. In order to overcome this disadvantage, Liu [2] introduces an estimation method for the camera parameters using prior knowledge about the distribution of relative human heights. Cho [6] proposes an estimation method for the camera parameters by tracking the poses of human body from a sequence of frames. The length estimation methods by the vanishing points use a principle that several parallel lines in 3D space meet at one point in a 2D image. The vanishing point is found by detecting the straight lines in the image. The length ratio between two objects can be calculated using the vanishing points. If the length of one object is known in advance, then the length of another object can be calculated by the length ratio. Criminisi [7] introduces a length estimation method by the given vanishing points of the ground. Fernanda [8] proposes the detection method of the vanishing points by clustering the straight lines iteratively without camera calibration. Jung [9] proposes the method of detecting the vanishing points for color videos captured by multiple cameras. Viswanath [10] proposes an error model for the human-height estimation by the vanishing points. The error of the human-height estimation is corrected by the error model. Rother [11] detects the vanishing points by tracking specific object such as traffic signs from a sequence of frames. Pribyl [12] estimates the object length by detecting the specific objects. Human height can be also estimated by the prior statistical knowledge of human anthropometry [28,29] or by the gaits [30,31]. In recent years, estimation studies in various fields achieve great success by applying neural networks. The neural networks are also applied to the human-height estimation. Gunel [32] proposes a neural network for predicting a relationship between each proportion of human joints and human height. Sayed [33] estimates human height by CNN using a length ratio between a human body width and a human head size.

Since depth video has distance information from the depth camera, the distance between two points in a depth image can be measured without the camera parameters or the vanishing points. Many studies [34–36] extract a skeleton, which is the connection structure of human body parts, from the depth image for the human-height estimation. However, the human body region extraction is some inaccurate due to noises in depth video.

## 3. Proposed Method

In this study, we propose a human-height estimation method using color and depth information. It is assumed that a depth camera is fixed in a certain position. Color and depth videos are captured by the depth camera. Then, a human body and a human head are extracted from current frame in color or depth video. A head-top and a foot-bottom are found in the human head and the human body, respectively. Two head-top and foot-bottom points are converted into 3D real-world coordinates by the corresponding pixel values of the frame in depth video. Human height is estimated by calculating a distance between two real-world coordinates. The flow of the proposed methods is shown in Figure 1.

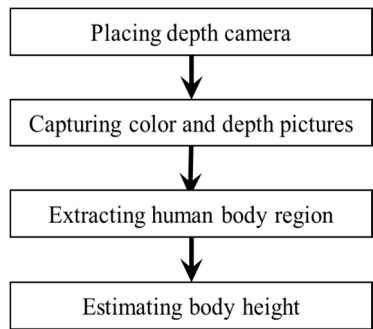

**Figure 1.** Flow of proposed method.

### 3.1. Human Body Region Extraction

It is important to accurately detect the human body region for estimating human height precisely. Most frames in depth video have many noises, so it is difficult to extract the accurate human body region from the depth frame. In contrast, color video allows to detect the human body region accurately by CNN. In the proposed method, mask R-CNN [26] is applied to extract the accurate human body region from current frame in color video. Then, the human body region is mapped to current frame in depth video. If color video is not available for extracting the human body region, then the human body region is extracted from current frame in depth video directly. In this case, the human body region is extracted by comparing with current depth frame with a pre-captured background depth image. Figure 2 shows the flow of extracting the human body region in the proposed method.

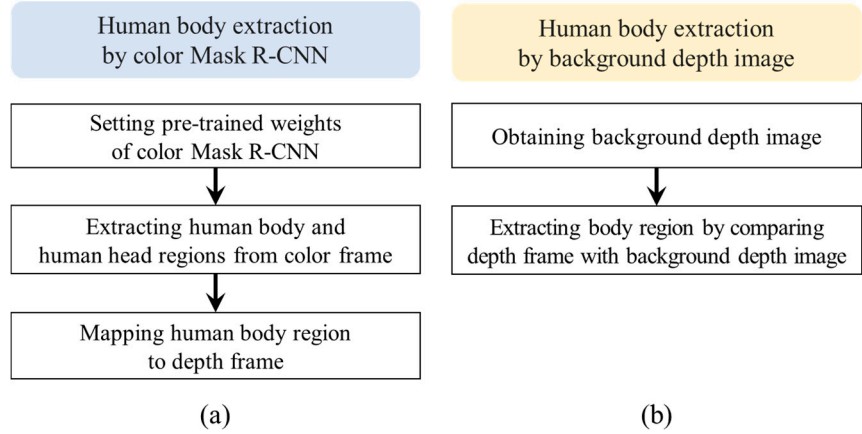

**Figure 2.** Flow of human body region extraction. (**a**) From color information; (**b**) from depth information.

### 3.1.1. Human Body Region Extraction Using Color Frames

Mask R-CNN [26] consists of three parts: a feature pyramid network (FPN) [37], a residual network (ResNet) [38,39] and a RPN. A FPN detects the categories and the boundary boxes of objects in color video. ResNet extracts an additional feature map from each boundary box. Figure 3 shows the processes of extracting the human body and the human head using mask R-CNN.

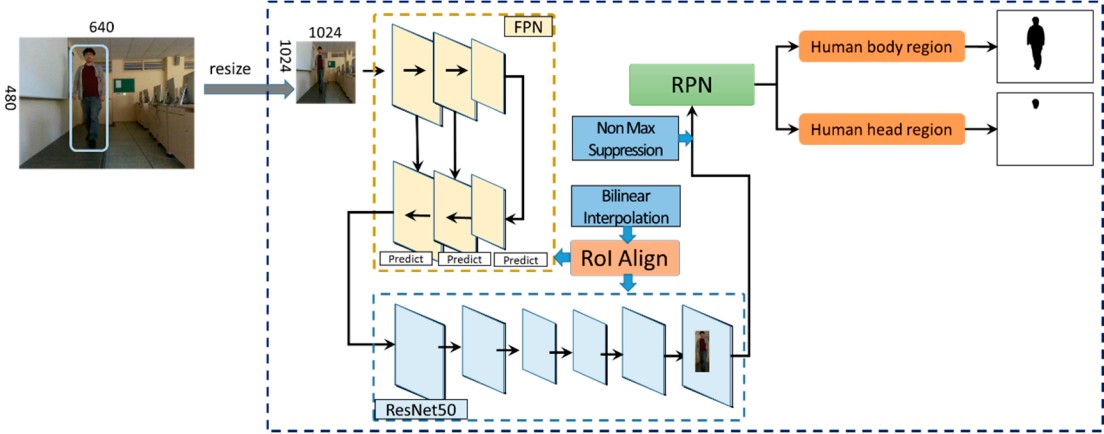

**Figure 3.** Processes of extracting human body and human head regions using mask R-CNN.

FPN condenses the scale of the input frame through bottom–up layers and expands the scale through top–down layers. Various scale objects can be detected through FPN. ResNet introduces a skip connection algorithm that the output value of each layer feeds into the next layer and directly into the layers about more than 2 hops away. The skip connection algorithm reduces the amount of values to be learned for weights of layers, so the learning efficiency of ResNet is improved. The feature map is extracted from the frame through FPN and ResNet. RPN extracts the boundary boxes and the masks which are object area in rectangle and in pixels, respectively. Comparing with faster R-CNN which applies RoIPool, mask R-CNN extends RPN to extract not only the boundary box, but also the masks. RoIPool rounds off the coordinates of the boundary box to integer. In contrast, RoIAlign allows the floating coordinates. Therefore, the detection of the object areas of mask R-CNN is more precisely than of faster R-CNN. Figure 4 shows an example for calculating the coordinates of the regions of interest (RoIs) for detecting the boundary box and the masks by RoIPool and RoIAlign. Non-max-suppression (NMS) removes overlapping areas between the boundary boxes. The size of the overlapping area is calculated for each boundary box. Two boundary boxes are merged if the size of the overlapping area is more than 70%.

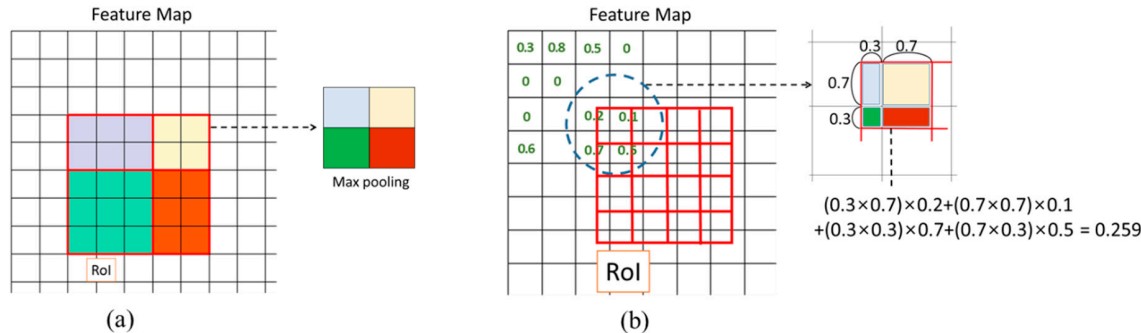

**Figure 4.** Example of region of interest (RoI) coordinate calculation by RoIPool and RoIAlign. (**a**) RoIPool; (**b**) RoIAlign.

Table 1 shows the performances of mask R-CNN when various types of backbones are applied to mask R-CNN. When X-101-FPN is applied as the backbone of mask R-CNN, the average precision of the boundary boxes (Box AP), which is a metric for detecting the boundary box, is the highest. However, the times for a train and a detection are slowest. In consideration of a tradeoff between the accuracy and the time for the detection, the proposed method applies ResNet-50 FPN which consists of 50 CNNs as the backbone.

**Table 1.** Performance of backbones for mask R-CNN. Bold represents the best performance.

| Backbone | Learn Rate Schedules | Train Time (s/iter) | Inference Time (s/im) | Box AP (%) |
|---|---|---|---|---|
| R50-C4 | 1x | 0.584 | 0.110 | 35.7 |
| R50-DC5 | 1x | 0.471 | 0.076 | 37.3 |
| ResNet-50 FPN | 1x | **0.261** | **0.043** | 37.9 |
| R50-C4 | 3x | 0.575 | 0.111 | 38.4 |
| R50-DC5 | 3x | 0.470 | 0.076 | 39.0 |
| ResNet-50 FPN | 3x | **0.261** | **0.043** | 40.2 |
| R101-C4 | 3x | 0.652 | 0.145 | 41.1 |
| R101-DC5 | 3x | 0.545 | 0.092 | 40.6 |
| ResNet-101 FPN | 3x | 0.340 | 0.056 | 42.0 |
| X-101-FPN | 3x | 0.690 | 0.103 | **43.0** |

The human body and the human head are detected by mask R-CNN. Mask R-CNN is trained using 3000 images of COCO dataset [40] with information about the human body and the human head. In the training mask R-CNN, a learn rate and epochs are set to 0.001 and 1000, respectively. A threshold for detection of the human body and the human head is set to 0.7. If a detection accuracy for RoI is more than the threshold, then corresponding RoI is detected as the human body or the human head. The process of extracting the human body and human head regions through mask R-CNN is as follows.

1. Resizing a color image to a certain size
2. Extracting a feature map through FPN and ResNet50
3. Extracting RoI boundary boxes from feature map by RoIAlign
4. Boundary box regression and classification for boundary boxes through RPN
5. Generating boundary box candidates by projecting the boundary box regression results onto the color frame
6. Detecting a boundary box for each object by non-max-suppression
7. Adjusting the boundary box area through RoIAlign
8. Finding pixels in boundary boxes to obtain a mask for each boundary box

### 3.1.2. Human Body Region Extraction Using Depth Frames

If the depth camera is fixed in a certain position, then the pixels of the human body region in the depth frame have different values from the depth pixels of a background. Therefore, the body region can be extracted by comparing depth pixels between the depth frame and the background depth image which has depth information about background. In order to extract the human body region accurately, the background depth image should be generated from several depth frames that capture the background because the depth video includes temporary noises. A depth value at the certain position of the background depth image is determined as a minimum value among pixels in the corresponding position of the depth frames capturing the background.

The human body region is extracted by comparing the pixels between the depth frame and the background depth image. A binarization image $B$ is generated for the human body region extraction as follows:

$$B(x,y) = \begin{cases} 1, & d_b(x,y) - d(x,y) > T_b \\ 0, & \text{otherwise} \end{cases}, \tag{1}$$

where $d_b(x,y)$ and $d(x,y)$ are the depth pixel values of the background depth image and the depth frame at position of $(x,y)$, respectively and $T_b$ is a threshold for the binarization.

### 3.2. Extraction of Head Top and Foot Bottom Points

The topmost point of the human head region is extracted as the head-top, $(x_h, y_h)$ and the bottommost point of the human body region as the foot-bottom, $(x_f, y_f)$. If horizontal continuous pixels

exist as shown in Figure 5, then the head-top or the foot-bottom is extracted as a center point among these pixels. If human stands with legs apart as shown in Figure 6, then two separate regions may be found from the bottommost of the human body region. In this case, the center points of two regions are the candidates of the foot-bottom. One candidate which has a depth value closer to the depth pixel value of the head-top point is selected as the foot-bottom point.

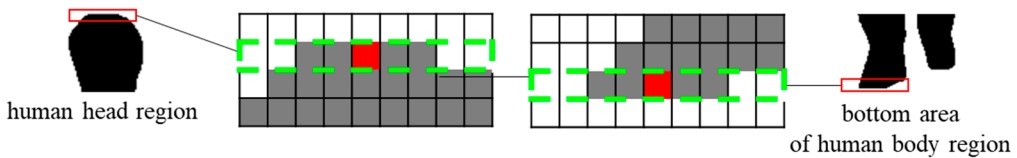

**Figure 5.** Extracting the head-top and foot-bottom points.

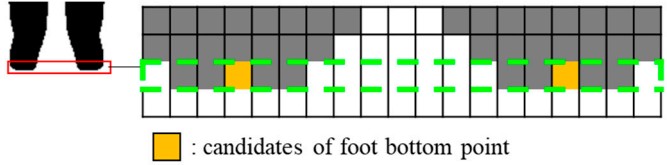

**Figure 6.** Extracting the foot-bottom point in case of apart human legs.

### 3.3. Human Height Estimation

Human height is estimated by measuring a length in the 3D real world between the head-top and foot-bottom points. In order to measure the length on the real world, 2D image coordinates of two head-top and the foot-bottom are converted into 3D real-world coordinates by applying a pinhole camera model [41] as follows:

$$
\begin{aligned}
X &= \frac{(x-W/2)}{f}d(x,y) \\
Y &= \frac{(y-H/2)}{f}d(x,y) \\
Z &= d(x,y),
\end{aligned}
\tag{2}
$$

where $X$, $Y$, $Z$ are the real-world coordinates, $f$ is a focal length of the depth camera, which means the parameter of the depth camera, and $W$ and $H$ are the horizontal and vertical resolutions of the depth image, respectively. In (2), the origin of the image coordinate system is the top–left of the image, but the origin of 3D camera coordinate system is the camera center. In order to compensate for the difference in the position of the origin between two coordinate systems, the coordinates of the image center are subtracted from the image coordinates. the real-world coordinates of the head-top ($X_h$, $Y_h$, $Z_h$) and of the foot-bottom ($X_f$, $Y_f$, $Z_f$) are calculated by substituting the real-world coordinates and the depth values of the head-top and the foot-bottom for (2), respectively, as follows:

$$
\begin{aligned}
X_h &= \frac{(x_h-W/2)}{f}d_h \\
Y_h &= \frac{(y_h-H/2)}{f}d_h \\
Z_h &= d_h,
\end{aligned}
\tag{3}
$$

$$
\begin{aligned}
X_f &= \frac{(x_f-W/2)}{f}d_f \\
Y_f &= \frac{(y_f-H/2)}{f}d_f \\
Z_f &= d_f,
\end{aligned}
\tag{4}
$$

where $d_h$ and $df$ are the depth values of the head-top and the foot-bottom, respectively. Human height is estimated by calculating an Euclidean distance between the real-world coordinates of the head-top and the foot-bottom as follows:

$$
\begin{aligned}
H &= \sqrt{\left(X_h - X_f\right)^2 + \left(Y_h - Y_f\right)^2 + \left(Z_h - Z_f\right)^2} \\
&= \sqrt{\left(\left(x_h d_h - x_f d_f\right)/f\right)^2 + \left(\left(y_h d_h - y_f d_f\right)/f\right)^2 + \left(d_h - d_f\right)^2}.
\end{aligned}
\tag{5}
$$

The unit of the estimated human height by (5) is same as the unit of the depth pixels. If the pixels of the depth video store the distance as millimeters, then the unit of $H$ is millimeter.

The estimated human height by (5) may have an error. One reason of the error is the noise of $d_h$. Generally, $(x_h, y_h)$ may be in a hair area. The depth values in the hair area have large noises because the hair causes the diffuse reflection of an infrared ray emitted by the depth camera. Therefore, $d_h$ should be corrected as the depth value of a point which is close to the head top but is not on the hair area. The depth value of the point which is not on the hair area has a high similarity to the depth values of neighboring pixels. The similarity is obtained by calculating the variance of the pixels located within $r$ pixels to the left, right and bottom including the corresponding pixel as follows:

$$
\sigma_r^2 = \frac{1}{(r+1)(2r+1)} \sum_{i=0}^{r} \sum_{j=-r}^{r} \left(d(x+i, y+j)^2\right) - \left(\frac{1}{(r+1)(2r+1)} \sum_{i=0}^{r} \sum_{j=-r}^{r} d(x+i, y+j)\right)^2.
\tag{6}
$$

If $\sigma_r^2$ is smaller than $T_\sigma$, then the $d_h$ is corrected as the depth value of the corresponding pixel as shown in Figure 7. Otherwise, the point is found between pixels below one pixel and the similarity of the found point is calculated by (6). In (6), $r$ is smaller as $d_h$ is larger because the width of an object is larger as the distance from the camera is closer as follows [42]:

$$
\frac{P_1}{P_2} = \frac{d_2}{d_1},
\tag{7}
$$

where $P_1$ and $P_2$ are the pixel lengths of the object widths when the depth values are $d_1$ and $d_2$, respectively. Therefore, $r$ depended on the depth value is determined as follows:

$$
r = \frac{d_0}{d_h} r_0.
\tag{8}
$$

In (8), $d_0$ and $r_0$ are constants so $d_0 \times r_0$ can be regarded as a parameter. If $d_0 \times r_0$ is represented as $\gamma$, (6) is modified as follows:

$$
\begin{aligned}
\sigma_r^2 &= \frac{1}{(\gamma/d_h+1)(2\gamma/d_h+1)} \sum_{i=0}^{\gamma/d_h} \sum_{j=-\gamma/d_h}^{\gamma/d_h} \left(d(x+i, y+j)^2\right) \\
&\quad - \left(\frac{1}{(\gamma/d_h+1)(2\gamma/d_h+1)} \sum_{i=0}^{2\gamma/d_h} \sum_{j=-2\gamma/d_h}^{2\gamma/d_h} d(x+i, y+j)\right)^2.
\end{aligned}
\tag{9}
$$

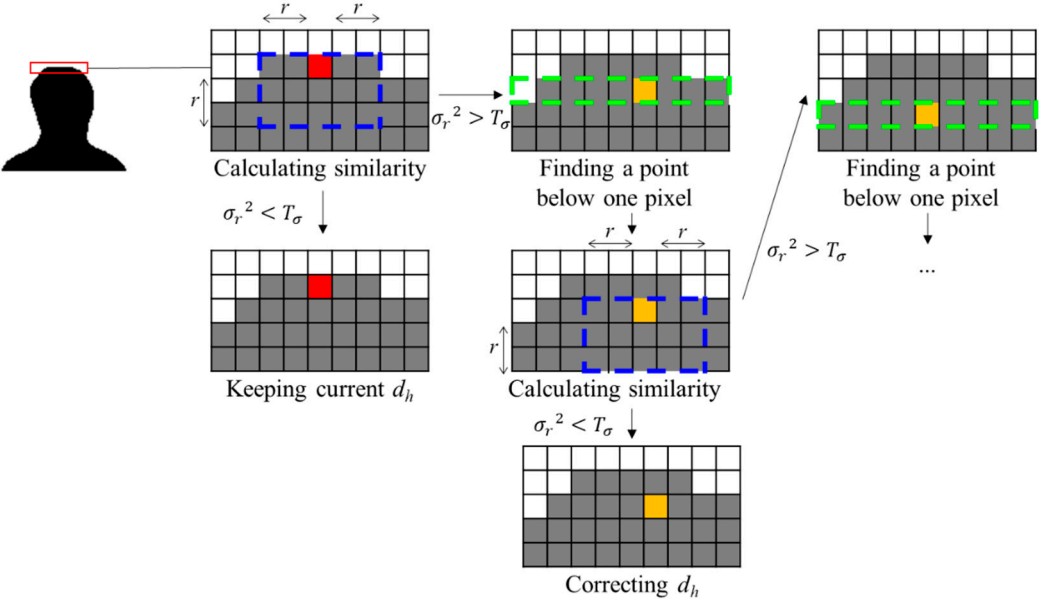

**Figure 7.** Flow of correcting $d_h$ by calculating similarity to neighboring pixels.

Mask R-CNN occasionally detects slightly wider human body region than the actual region. In particular, the region detection error in the lower part of human body may causes that a point on the ground is extracted as the foot-bottom. Assuming the ground is flat, the difference in a depth gradient is little between two vertically adjacent pixels that is in the ground area. The depth gradient of certain pixel $(x, y)$ is defined as follows:

$$g(x, y) = d(x + 1, y) - d(x, y). \tag{10}$$

If certain point is on the ground, then the difference in the depth gradients is the same between the point and one pixel down. In order to determine whether the extraction of the foot-bottom is correct, two depth gradients are compared as follows:

$$
\begin{aligned}
D &= g(x_f - 1, y_f) - g(x_f, y_f) \\
&= \left( d(x_f, y_f) - d(x_f - 1, y_f) \right) - \left( d(x_f + 1, y_f) - d(x_f, y_f) \right) \\
&= 2d(x_f, y_f) - d(x_f - 1, y_f) - d(x_f + 1, y_f).
\end{aligned}
\tag{11}
$$

If $D$ is 0, then the point is removed from the human body region. The comparison of the depth gradients by (11) is applied to the bottommost pixels of the human body region in order to correct the foot-bottom. If all of the bottommost pixels are removed, then this process is repeated for the points of the human body region where is one pixel up. The foot-bottom is extracted as a center pixel among the bottommost pixels which are not removed. Figure 8 shows correcting the position of the foot-bottom point.

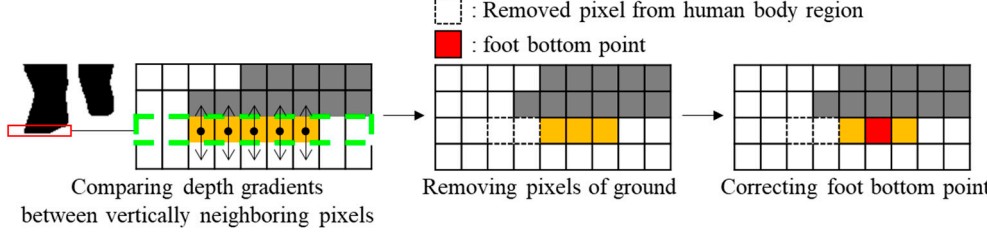

**Figure 8.** Foot bottom point correction.

The noises of depth video may cause the temporary error of the human-height estimation. The temporary error of the human height is corrected by the average of the estimated human heights among a sequence of depth frames as follows:

$$
\begin{aligned}
\overline{H}(n+1) &= \frac{1}{n+1}\left(\sum_{i}^{n+1} H(i)\right) \\
&= \frac{1}{n}\left(\sum_{i}^{n} H(i)\right) \times \frac{n}{n+1} + \frac{1}{n+1}H(n+1) \\
&= \frac{n}{n+1}\overline{H}(n) + \frac{1}{n+1}H(n+1),
\end{aligned}
\tag{12}
$$

where $n$ is the order of the captured depth frames and $H(n)$ and $\overline{H}(n)$ are the estimated and corrected human heights in the $n$th frame order, respectively.

## 4. Experiment Results

Intel Realsense D435 is used as a depth camera for the experiments of the proposed method. A focal length $f$ and a frame rate of the depth camera are 325.8 mm and 30 Hz, respectively. The resolutions of depth video are specified as 640 × 480. The threshold $T_b$ for (1) and $T_\sigma$ for (6) are set to 100 and 50, respectively. The parameter $\gamma$ in (9) is 4000, which means $r$ is 2 when $d_h$ is 2000 mm.

Figures 9 and 10 show the extractions of the human body region through by mask R-CNN and by the background depth image, respectively. Both methods of the human body region extraction accurately extract the human body region at not only a standing state, but also a walking state. In addition, the human body region is accurately extracted regardless of the states of the human body. In Figure 9, areas painted in green and red are the human body and human head regions, respectively. The human head regions are accurately found even though the position of the hand is above the head. The human body region extraction by the background depth image extracts larger regions than by mask R-CNN, so some part of the background is included in the human body region. In addition, the bottom area of the human body is not included in the human body region because the depth values of these area are similar to the depth value of the ground.

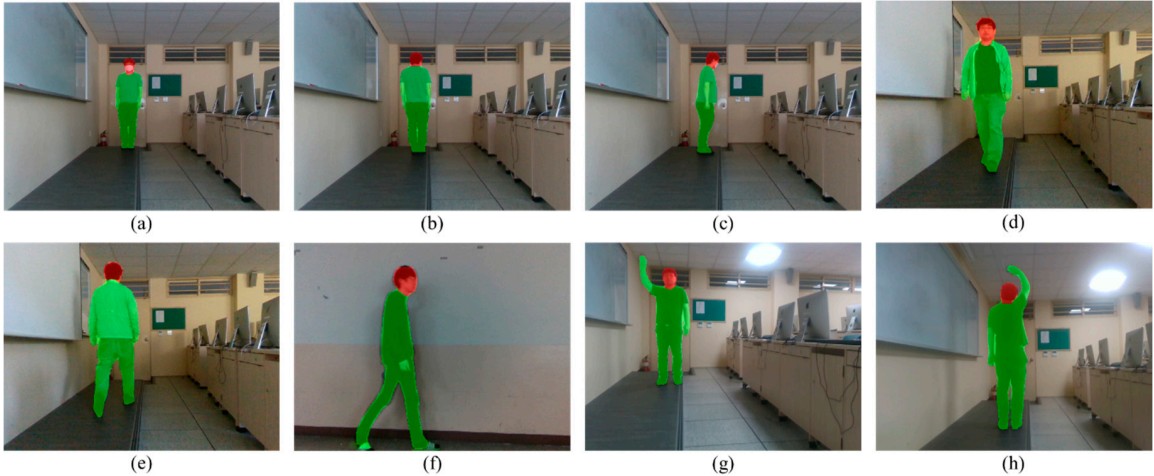

**Figure 9.** Extraction of human body region by mask R-CNN. (**a**) Standing toward front; (**b**) standing backward; (**c**) standing sideways; (**d**) walking toward camera; (**e**) walking opposite to camera; (**f**) lateral walking, (**g**) standing toward front and waving hand; (**h**) standing backward and waving hand.

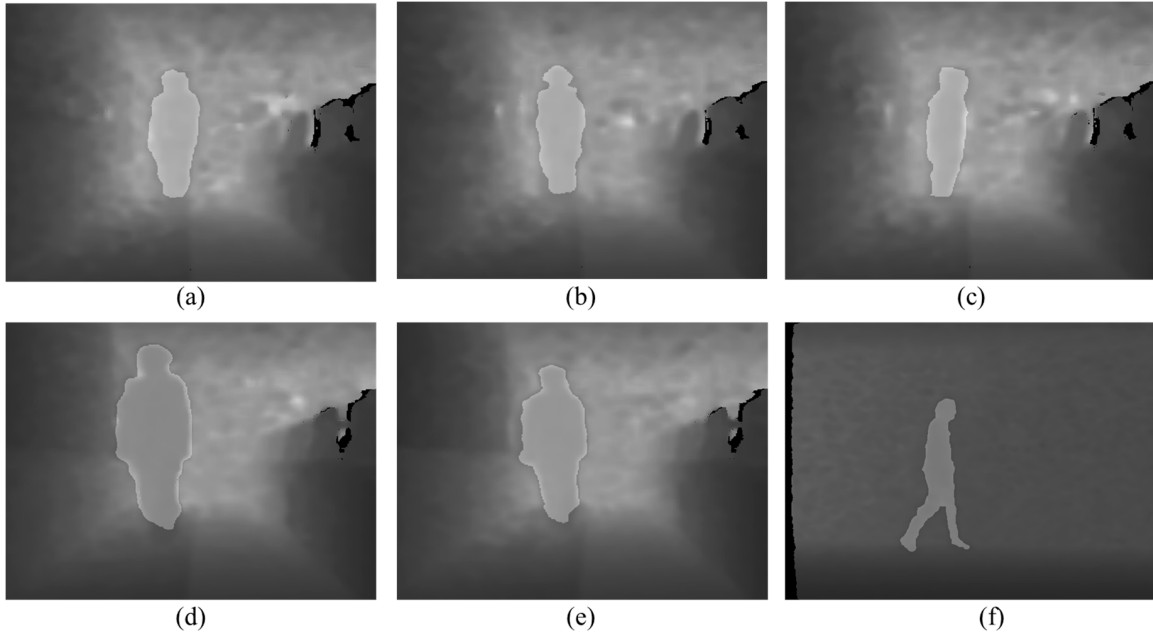

**Figure 10.** Extraction of human body region by background depth image. (**a**) Standing toward front; (**b**) standing backward; (**c**) standing sideways; (**d**) walking toward camera; (**e**) walking opposite to camera; (**f**) lateral walking.

Figure 11 shows the correction of human height by (12) when the body region is extracted by mask R-CNN. The distributions of the estimated human height are large because of the noises of the depth frame when the correction of human height is not applied. After applying the correction of the human-height estimation by (12), the human heights are estimated as certain heights after about 20 frames.

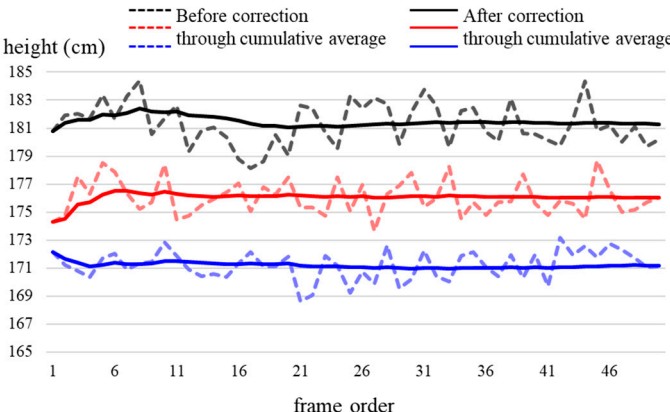

**Figure 11.** Results of height estimation after correction through cumulative average for each three persons.

Figure 12 shows the result of the human-height estimation depending on the methods of the human body region extraction. The actual height of a person is 177 cm. In the human body region extraction through the background depth image, first 50 frames are accumulated to generate the background depth image. The human body keeps at a distance of 3.5 m from the camera. The body height is estimated as 176.2 cm when the human body region is extracted by mask R-CNN. The body height is estimated as 172.9 cm when the human body region is extracted by the background depth image.

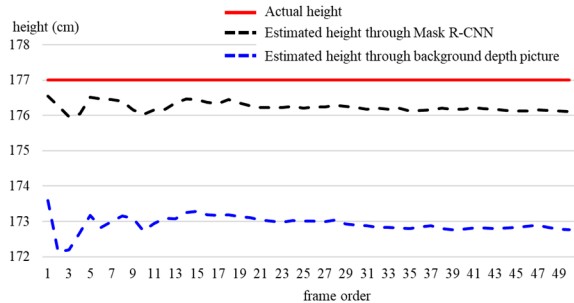

**Figure 12.** Results of height estimation depending on methods of human body region extraction.

Figure 13 shows the result of the human-height estimation according to the distance between human body and the camera. The actual height of a person is 182 cm. The distances from the camera are 2.5 m, 3 m, 3.5 m, 4 m and 4.5 m. The averages of the human heights are estimated as 181.5 cm, 181.1 cm, 181.2 cm, 179.7 cm and 179.8 cm when the distances are 2.5 m, 3 m, 3.5 m, 4 m and 4.5 m, respectively.

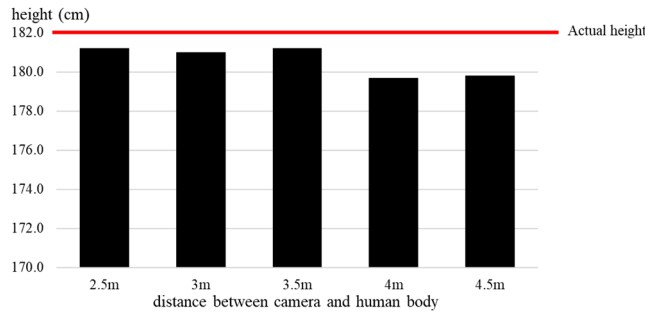

**Figure 13.** Results of height estimation according to distance between camera and human body.

Figure 14 shows the result of the human-height estimation when a person whose height is 180 cm is standing, walking toward the camera and lateral walking. Human body keeps at a distance of 2.5 m from the camera when the human is standing and lateral walking. When the human is walking toward the camera, the distance from the camera is in range of 2.5 m to 4 m. In the standing state, the human height is estimated as 178.9 m. The human height is 177.1 cm and 174.9 cm in the lateral walking and the walking toward the camera, respectively. The magnitude of the estimated error in the lateral walking state is similar to in the standing state. The estimated error in walking toward the camera is larger than the others. The reason is that the vertical length of the human body is reduced because human knees are bent to a certain degree while walking.

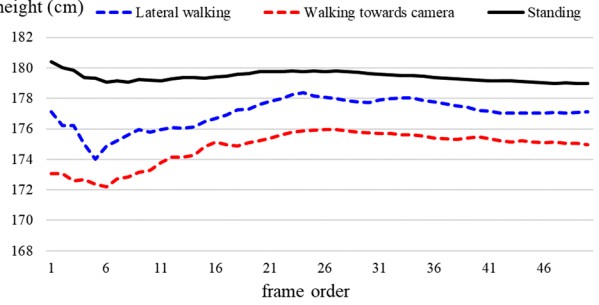

**Figure 14.** Results of height estimation in standing and walking states.

Figure 15 shows the positions of $d_h$ and $(x_f, y_f)$ before and after the correction of the head-top and the foot-bottom, respectively. The green and red points in Figure 15 represent the head-top and

foot-bottom points, respectively. In Figure 15a, the position of $d_h$ is on the hair area, so $d_h$ has some error and the changes in $d_h$ are large as shown in Figure 16. After correcting $d_h$, the changes is smaller. The changes of the estimated human body height are reduced after correcting of the foot-bottom point as shown in Figure 17. Two persons whose actual heights are 182 cm and 165 cm are estimated as 188.6 cm and 181.5 cm before correcting $d_h$, respectively, as 172.2 cm and 163.3 cm after the correction of the head-top point, respectively and as 181.5 cm and 163.1 cm after the correction of both head-top and foot-bottom points, respectively.

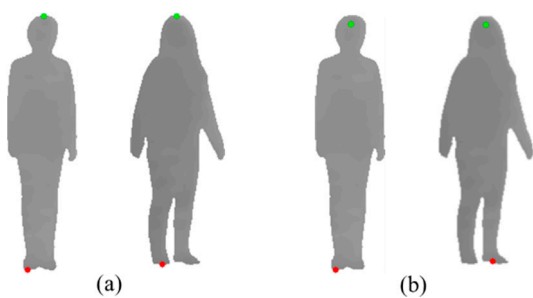

**Figure 15.** Positions of $d_h$ and $(x_f, y_f)$. (**a**) Before correction; (**b**) after correction.

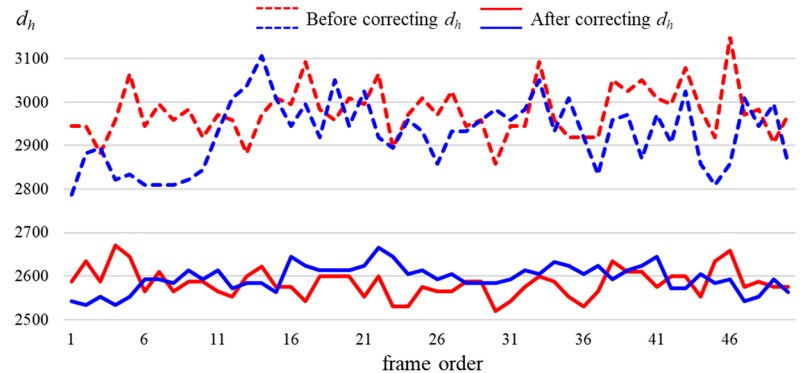

**Figure 16.** Changes in $d_h$ according to frame order.

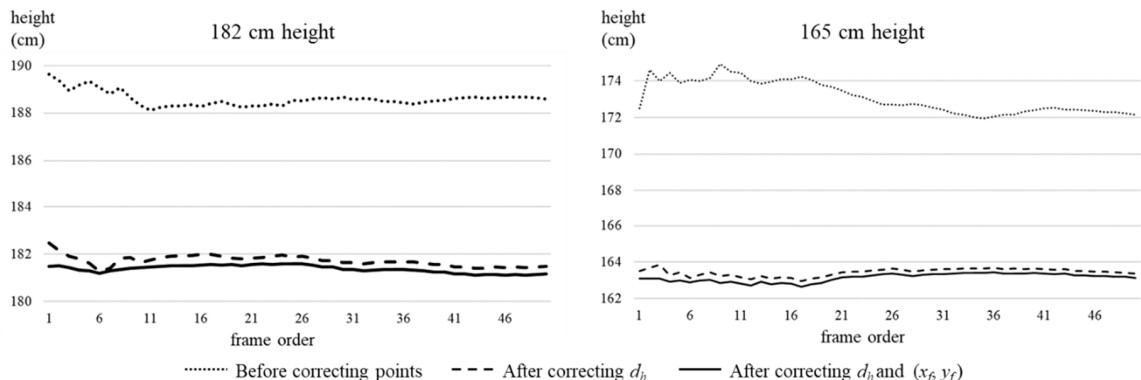

**Figure 17.** Result of height estimation after correction of $d_h$ and $(x_f, y_f)$.

Figure 18 shows the results of the human-height estimation depending on $r_0$ and $T_\sigma$, which are the parameters for (8) and (9) when $d_0$ is 2000. The estimated height drops sharply when $r_0$ is less than or equal to 2 and decreases smoothly when $r_0$ is larger than 2. In addition, the estimated height linearly increases when $T_\sigma$ is less than or equal to 250 and slowly increases when $T_\sigma$ is larger than 250. Body height is estimated most accurately when $r_0$ is 2 and $T_\sigma$ is 125.

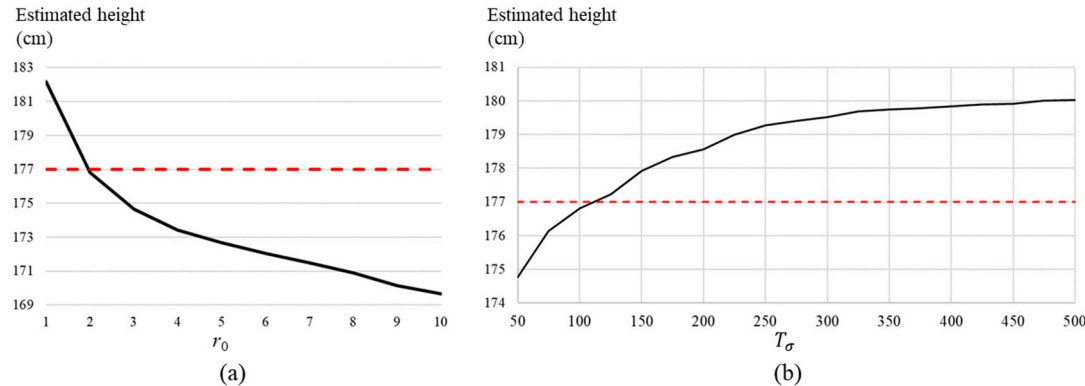

**Figure 18.** Results of human-height estimation depending on parameters for (8) and (9). (**a**) *r*; (**b**) $T_\sigma$.

Tables 2–4 show the results of the human-height estimation depending on human body postures for 10 persons. All of persons are captured within a range of 2.5 m to 4 m from the camera. Each person is captured with 150 frames. The error of the human-height estimation is calculated as follows:

$$\frac{\left|\overline{H} - H_{actual}\right|}{H_{actual}}, \tag{13}$$

where $\overline{H}$ and $H_{actual}$ are the average of the corrected human heights by (12) and an actual height for a person, respectively. When the human body region is extracted by mask R-CNN, the errors of the human-height estimation with standing, lateral walking and walking towards the camera are 0.7%, 1.3% and 1.8%, respectively. The accurate foot-bottom for the human-height estimation is the point of a foot heel which is on the ground. However, the bottommost pixel of the body region which is extracted as the foot-bottom point is usually a foot toe point in the proposed method. The position difference between the foot heel and foot toe points may make the error of the human-height estimation. The human-height estimation errors with standing, lateral walking, and walking towards the camera are 2.2%, 2.9%, 4.6%, respectively, when the body region is extracted by the background depth image. The human-height estimation using only depth frames has more error than using both color and depth frames.

**Table 2.** Results of human-height estimation by proposed method while standing.

| Person No. | Actual Height (cm) | Extracting Human Body by Mask R-CNN | | Extracting Human Body by Background Depth Image | |
|---|---|---|---|---|---|
| | | Estimated Height (cm) | Estimation Error (%) | Estimated Height (cm) | Estimation Error (%) |
| 1 | 177 | 179.6 | 1.5 | 172.1 | 2.8 |
| 2 | 183 | 182.1 | 0.5 | 176.5 | 3.6 |
| 3 | 165 | 164.2 | 0.5 | 161.3 | 2.2 |
| 4 | 178 | 176.5 | 0.8 | 173.9 | 2.3 |
| 5 | 182 | 180.9 | 0.6 | 177.7 | 2.4 |
| 6 | 173 | 174.6 | 0.9 | 175.2 | 1.3 |
| 7 | 175 | 174.4 | 0.3 | 171.3 | 2.1 |
| 8 | 170 | 169.2 | 0.5 | 167.1 | 1.7 |
| 9 | 168 | 167.3 | 0.4 | 165.6 | 1.4 |
| 10 | 181 | 178.9 | 0.6 | 177.4 | 1.4 |
| Average error (%) | | 0.7 | | 2.2 | |

**Table 3.** Results of human-height estimation by proposed method while lateral walking.

| Person No. | Actual Height (cm) | Extracting Human Body by Mask R-CNN | | Extracting Human Body by Background Depth Image | |
|:---:|:---:|:---:|:---:|:---:|:---:|
| | | Estimated Height (cm) | Estimation Error (%) | Estimated Height (cm) | Estimation Error (%) |
| 1 | 177 | 176.1 | 0.5 | 171.5 | 3.1 |
| 2 | 183 | 180.7 | 1.3 | 175.9 | 3.9 |
| 3 | 165 | 163.0 | 1.2 | 160.1 | 3.0 |
| 4 | 178 | 175.2 | 1.6 | 174.1 | 2.2 |
| 5 | 182 | 179.6 | 1.3 | 176.6 | 3.0 |
| 6 | 173 | 170.7 | 1.3 | 168.1 | 2.8 |
| 7 | 175 | 172.6 | 1.4 | 170.7 | 2.5 |
| 8 | 170 | 167.5 | 1.5 | 166.1 | 2.3 |
| 9 | 168 | 166.2 | 1.1 | 162.8 | 3.1 |
| 10 | 181 | 177.1 | 1.6 | 174.8 | 2.9 |
| Average error (%) | | 1.3 | | 2.9 | |

**Table 4.** Results of human-height estimation by proposed method while walking towards camera.

| Person No. | Actual Height (cm) | Extracting Human Body by Mask R-CNN | | Extracting Human Body by Background Depth Image | |
|:---:|:---:|:---:|:---:|:---:|:---:|
| | | Estimated Height (cm) | Estimation Error (%) | Estimated Height (cm) | Estimation Error (%) |
| 1 | 177 | 174.1 | 1.6 | 167.5 | 5.4 |
| 2 | 183 | 178.7 | 2.3 | 172.9 | 5.5 |
| 3 | 165 | 161.0 | 2.4 | 157.1 | 4.8 |
| 4 | 178 | 173.8 | 2.4 | 172.1 | 3.3 |
| 5 | 182 | 178.4 | 2.0 | 173.6 | 4.6 |
| 6 | 173 | 169.7 | 1.9 | 164.1 | 5.1 |
| 7 | 175 | 171.2 | 2.2 | 168.7 | 3.6 |
| 8 | 170 | 166.4 | 2.1 | 163.1 | 4.1 |
| 9 | 168 | 165.2 | 1.7 | 158.8 | 5.5 |
| 10 | 181 | 174.9 | 2.8 | 171.8 | 4.6 |
| Average error (%) | | 2.1 | | 4.6 | |

## 5. Conclusions

In this study, a human-height estimation method using color and depth information was proposed. The human body region was extracted through the pre-trained mask R-CNN to color video. The human body region extraction from depth video was also proposed by comparing with the background depth image. Human height was estimated from depth information by converting two points of head-top and foot-bottom into two 3D real-world coordinates and by measuring the Euclidean distance between two 3D coordinates. Human height was accurately estimated even if the person is not in front or a walking state. In the experiment results, the errors of the human-height estimation by the proposed method with the standing state were 0.7% and 2.2% when the human body region was extracted by mask-R CNN and by the background depth image, respectively. The proposed method significantly improves the human-height estimation by combining color and depth information. The proposed method can be applied to estimate not only the body height, but also the height of other object types such as animals. The proposed method can also be applied to gesture recognition and body posture estimation which require the types and the 3D information of objects.

**Author Contributions:** Conceptualization, D.-s.L., J.-s.K., S.C.J. and S.-k.K.; software, D.-s.L. and J.-s.K.; writing—original draft preparation, D.-s.L., J.-s.K., S.C.J. and S.-k.K.; supervision, S.-k.K. All authors have read and agreed to the published version of the manuscript.

**Funding:** This research was supported by the BB21+ Project in 2020 and was supported by Dong-eui University Foundation Grant (2020).

**Conflicts of Interest:** The authors declare no conflict of interest.

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
