# Peer review of "Human Height Estimation by Color Deep Learning and Depth 3D Conversion"

_applsci, doi:10.3390/app10165531_

Round 1

Reviewer 1 Report

Please explain how you defined the weights 0.2, 0.5, 0.1, 0.2 in the formula at the end of line 169.

The heel of the human foot and the top of the head determine the height when the foot and body are straight. In line 199, in the image taken from the front, the front of the foot may not be suitable for this, because the front of the foot (first toes) is on the ground even when the heel is raised from the ground and the foot is not straight. But, of course, it fits an approximation.

Author Response

Point 1: Please explain how you defined the weights 0.2, 0.5, 0.1, 0.2 in the formula at the end of line 169.

Response 1: We modified the figure and its captions of Fig. 4 to describe the weights (Line 178).

Point 2: The heel of the human foot and the top of the head determine the height when the foot and body are straight. In line 199, in the image taken from the front, the front of the foot may not be suitable for this, because the front of the foot (first toes) is on the ground even when the heel is raised from the ground and the foot is not straight. But, of course, it fits an approximation.

Response 2: It is correct to estimate the human height by measuring the distance of the head crown and the foot heel. However, the heel is often hidden in other body parts when the body stands frontal. It makes detecting the heel difficult in the image so we detect the body floor point as the bottom-most point of the human body region. At the results, the accuracy of the human height estimation is decreased. We added this discussion in the experimental results section as follows. 

  • (Line 367) The accurate body floor for the human height estimation is the point of a foot heel which is on the ground. However, the bottom-most pixel of the body region which is extracted as the body floor point in the proposed method is usually a foot toe point. The position difference between the foot heel and the foot toe points may make the error of the human height estimation.

Reviewer 2 Report

This paper proposed human height estimation method using the color and depth information.
There are a lot of research works that have been done for estimating the human height using image or video data.
Therefore, it is very important to explain about the contributions and differences of the proposed method with the other methods.
The author should clearly explain about the advantages of the proposed method in compare with the other methods.

The following are some comments for modifying the paper to become as a contributed research paper form.

1. The abstract is very simple and need to describe the contributions of the proposed system to be more obvious and impressive.

2. Section names need to be clearly described. For example:
Section 3 Proposed Method
Section 3.1 Human body region extraction
3.1.1 Human body region extraction using color frames
3.1.2 Human body region extraction using depth frames

3. Should rewrite the equation of generating binary mask using thresholding method by adding the conditions.

4. In image processing and computer vision research, the word "picture" is not use and instead use the words such as "image" or "frame".

5. Before the section of "Human height estimation", the author should explain about the "Extraction of head top and body floor points" as a section.

6. In that section, the author should also explain about results of correction of head top point and how to find the head region point at the time of hands location is above the head region (for example: Waving hand action).

7. In that section, the author should also explain about how to handle region which is located near the foot position and results of correction of foot floor point.

8. In the experimental results section, the author need toe show the results of human height estimation by proposed method in actions of standing, walking toward the camera, and lateral walking because depending on the action, the error of height estimation error % will be different. (Should include three tables about the results of human height estimation by proposed method.)

9. Should perform more experiments to confirm the effectiveness of the proposed methods.

Author Response

Point 1: The abstract is very simple and need to describe the contributions of the proposed system to be more obvious and impressive.

 Response 1: We modified the abstract to describe the human height estimation more detail as follows.

  • (Line 12) In this paper, the estimation method for human height is proposed using color and depth information. Mask R-CNN is applied to the detection of a human body and a human head regions separately in a color image. If the color video is not available for extracting the human body region due to low light environment, then the human body region is extracted by comparing between a frame in depth video and a background depth image. Two points of a top head and a body floor are extracted as the top-most point of the human head region and as the bottom-most point of the human body region, respectively. The depth value of the head top point is corrected to a pixel value where has high similarity from a neighboring pixel. The position of the body floor point is corrected by calculating the depth gradient between vertically adjacent pixels. Two points are converted into 3D real world coordinates using depth information. Human height is estimated by measuring a Euclidean distance between two real world coordinates. The estimation errors for human height are corrected as the average of accumulated heights. In experiment results, we achieve that the estimated errors of human height with a standing state are 0.7% and 2.2% when the human body region is extracted by Mask R-CNN and the background depth image, respectively.

Point 2: Section names need to be clearly described.

 Response 2: The selection titles are modified as follows.

  • Section 3. Proposed Method
    • Section 3.1. Human body region extraction
      • Section 3.1.1. Human body region extraction using color frames
      • Section 3.1.2. Human body region extraction using depth frames
    • Section 3.2. Extraction of head top and body floor points
    • Section 3.3. Human height estimation

Point 3: Should rewrite the equation of generating binary mask using thresholding method by adding the conditions.

Response 3: We modified the equation of generating binary mask in line 210.

Point 4: In image processing and computer vision research, the word "picture" is not use and instead use the words such as "image" or "frame".

Response 4: We changed the term ‘picture’ to ‘image’ or ‘frame’.

Point 5: Before the section of "Human height estimation", the author should explain about the "Extraction of head top and body floor points" as a section.

Response 5: We added the section of ‘Extraction of head top and body floor points’ as follows.

  • (Line 213) The head top and the body floor points, (xh, yh) and (xf, yf), are extracted as the top-most pixel of the human head region and the bottom-most pixel of the human body region, respectively. If horizontally multiple pixels exist as shown in Fig. 5, then the head top or the body floor point is extracted as a center point among these pixels. If the human stands with legs apart as shown in Fig. 6, then two separate regions may be found from the bottom-most pixels. In this case, the center points of two regions are the candidates of the body floor point. One candidate of the body floor point which has a depth value closer to the depth pixel value of the head top point is selected as the body floor point for more accurate human height estimation.

Point 6: In that section, the author should also explain about results of correction of head top point and how to find the head region point at the time of hands location is above the head region (for example: Waving hand action).

Response 6: We modified Mask R-CNN to detect a human body and a human head regions separately in order to accurately extract the head top point even though the other part of the human body is above the head.

  • (Line 192) The human body and the human head regions are detected by Mask R-CNN. Mask R-CNN is trained using 3000 images of COCO dataset [40] with information about the human body and the human head regions.

 We modified that the head top point is extracted from the human head region.

  • (Line 214) The head top and the body floor points, (xh, yh) and (xf, yf), are extracted as the top-most pixel of the human head region and the bottom-most pixel of the human body region, respectively.

We added the results of the human head detection when the hand is above the head as follows.

  • (Line 289) In Fig. 9, areas painted in green and red are the human body and the human head regions, respectively. The human head regions are accurately found even though the location of the hand is above the head.

Point 7: In that section, the author should also explain about how to handle region which is located near the foot position and results of correction of foot floor point.

Response 7: Depending on the accuracy of detecting the human body region, the detected body floor point may include the ground close to the human body. In order to extract the accurate body floor point, we determined whether the body floor point is in the actual human body region by calculating the depth gradient between neighboring pixels.

  • (Line 263) The human body region occasionally includes adjacent ground area. In order to determine whether the body floor point is correct, the depth pixel values of vertically neighboring pixels are compared. If the extracted body floor point is in the ground area, then the depth gradient of a below pixel that is defined as a depth pixel difference is equal to the depth gradient of a above pixel. The difference of the depth gradients is calculated as follows.

  If the depth gradients of neighboring pixels are not different, then the corresponding pixel is assumed as the ground area and is removed from the human body region. The comparison of the depth gradients by (10) is applied to the bottom-most pixels of the human body region. If all of the bottom-most pixels are removed, then this process is repeated for the pixels just above. The body floor point is extracted as a center pixel among the bottom-most pixels which are not removed. Fig. 8 shows correcting the position of the body floor point.

Figure 8. Body floor point correction.

In addition, the results and the discussion of the body floor point correction are added in the section of the experiment results as follows. 

  • (Line 341) The difference in location of the body floor points is small before and after the correction. However, the changes of the estimated human body height are reduced after correcting of the body floor point as shown in Fig. 17.

Round 2

Reviewer 2 Report

The authors have revised the detail explanation of the paper according to my previous comments and suggestion.

Therefore, I agree to publish this paper in Applied Sciences journal after checking the English grammar and sentence structure.

Author Response

We would like to express our appreciations for your repeated review.

Point 1: The authors have revised the detail explanation of the paper according to my previous comments and suggestion.Therefore, I agree to publish this paper in Applied Sciences journal after checking the English grammar and sentence structure.

Response 1: We corrected English grammar and sentence structure.

We would like to appreciate your detailed review again.